# Probing the Knowledge Boundary:
# An Interactive Agentic Framework for Deep Knowledge Extraction

Yuheng Yang [1 2]   Siqi Zhu [1]   Tao Feng [1]   Ge Liu [1]   Jiaxuan You [1]

## Abstract

Large Language Models (LLMs) can be seen as compressed knowledge bases, but it remains unclear what knowledge they truly contain and how far their knowledge boundary extends. Existing benchmarks are mostly static and provide limited support for systematic knowledge probing. In this paper, we propose an interactive agentic framework to systematically extract and quantify the knowledge of LLMs. Our method includes four adaptive exploration policies to probe knowledge at different granularity. To ensure the quality of extracted knowledge, we introduce a three-stage knowledge processing pipeline that combines vector-based filtering to remove strict duplicates, LLM-based adjudication to resolve ambiguous semantic overlap, and domain relevance auditing to retain valid knowledge units. Through extensive experiments, we find that Recursive Taxonomy is the most effective exploration strategy. We also observe a clear knowledge scaling law, where larger models consistently recover more knowledge. In addition, we identify a Pass@1 versus Pass@k trade-off: domain-specialized models achieve higher initial accuracy but experience rapid degradation, while general-purpose models maintain stable performance over extended extraction. Finally, our results show that differences in training data composition lead to distinct and measurable knowledge profiles across model families, reflecting how pretraining shapes each model's parametric knowledge.

[1]University of Illinois at Urbana-Champaign, Urbana, IL, USA [2]Westlake University, Hangzhou, China. Correspondence to: Yuheng Yang <yangyuheng@westlake.edu.cn>, Jiaxuan You <jiaxuan@illinois.edu>.

*Proceedings of the 43[rd] International Conference on Machine Learning*, Seoul, South Korea. PMLR 306, 2026. Copyright 2026 by the author(s).

## 1. Introduction

Large Language Models (LLMs) have established themselves as the de facto engines of general-purpose knowledge, exhibiting capabilities that span a vast array of topics. As these models are increasingly integrated into high-stakes domains such as scientific research and decision support systems, understanding their behavior extends beyond simple performance metrics. It becomes imperative to explicitly characterize and measure the *knowledge boundary* of a black-box LLM — or more precisely, its *extractable knowledge frontier*: the set of unique, valid knowledge atoms that can be retrieved under a fixed interactive pipeline (formalized in §3). Delineating this boundary is critical for interpretability, alignment, and safety; without it, we cannot distinguish between a model accessing its latent knowledge and one fabricating information via hallucination, nor can we reliably predict failure modes in deployment.

However, quantifying this boundary presents three fundamental challenges that prevent the problem from being reduced to traditional accuracy evaluation. First, the definition of a "knowledge ceiling" lacks a verifiable ground truth in open-ended domains, rendering standard recall metrics inapplicable. Recent work has begun to investigate how LLMs perceive their own knowledge boundaries through internal representations (Xiao et al., 2025; Ni et al., 2025), but these approaches require white-box access. Second, current evaluation paradigms rely heavily on static benchmarks (e.g., MMLU) and predefined question sets (Hendrycks et al., 2021). These approaches not only suffer from increasing risks of data contamination but are also inherently passive; they sample the model's capabilities at sparse, discrete points (Ghosh et al., 2025) rather than actively exploring the continuous landscape of its knowledge coverage. Third, and perhaps most critically, LLMs exhibit a strong inertia to remain within a "comfort zone" of high-probability tokens. When queried directly, models tend to output generic or repetitive shallow knowledge. This behavior, coupled with the diversity of semantic expressions for identical facts, makes it profoundly difficult to achieve "saturation-based" extraction, where one must distinguish unique novel insights from semantic redundancies and hallucinations.

Prior literature has largely failed to address the systemic

nature of these challenges. Existing works predominantly focus on static benchmark-driven evaluations or employ single-strategy agentic extraction that explores the model locally. Such settings lack the adaptive, hierarchical, and recursive mechanisms necessary to push the model to its limits. Consequently, current methods often measure what the model is willing to express (surface realization), rather than approximating what the model actually knows (latent capacity). There remains a significant gap in methodologies that model the "merging and verification" of knowledge points as a central component of the evaluation process.

To bridge this gap, we propose an **Interactive Agentic Framework for Knowledge Extraction and Evaluation**, designed to systematically answer: What does the model know? How much does it know? And how can we maximize the excavation of this knowledge? Our framework treats the black-box model as an environment to be explored, deploying multiple **Agentic Extraction Pipelines** to break through the model's output inertia. Crucially, these pipelines are coupled with a robust **Knowledge Processor**, which handles the semantic de-duplication, validity verification, and structural organization of the generated content. Among our proposed strategies, the RECURSIVE TAXONOMY strategy demonstrates the highest efficacy in coverage and efficiency.

Our contributions are as follows:

- We provide a formal problem formulation for black-box knowledge boundary probing that moves beyond static benchmark testing toward dynamic, saturation-based interactive evaluation.

- We introduce a novel multi-agent framework that adaptively navigates the model's knowledge space, significantly outperforming naive prompting and standard chain-of-thought baselines in uncovering deep and long-tail knowledge.

- Empirically, we conduct four systematic experiments to characterize the knowledge topology of black-box LLMs. We first establish that *Recursive Taxonomy* is the Pareto-optimal extraction strategy, significantly outperforming naive probing methods. We then identify a *Knowledge Scaling Law*: extractable knowledge grows with model size, with larger models achieving superior coverage of long-tail concepts. We also reveal a *Pass@1 vs Pass@k trade-off*: domain-specific fine-tuning improves initial accuracy but causes rapid degradation and reduced recall. Finally, we demonstrate that *training data composition* dominates knowledge quality, as different model families exhibit stable recall patterns but domain-specific accuracy variations.

**Conflict of Interest Disclosure.** This work evaluates several commercial and open-source LLMs, including models from Meta (Llama-3.1), Alibaba (Qwen-2.5), and DeepSeek. None of the authors have a financial relationship with, or hold equity in, any of these organizations. The choice of models was made solely on the basis of public availability and relevance to the research questions.

## 2. Related Work

Our work intersects multiple research directions: knowledge evaluation in LLMs, agentic systems for exploration, and semantic deduplication. We position our contributions relative to each area below.

**Knowledge Evaluation in LLMs.** Traditional approaches rely on static benchmarks such as MMLU (Hendrycks et al., 2021) and TruthfulQA (Lin et al., 2022), which suffer from data contamination risk (Magar & Schwartz, 2022), passive sampling of sparse knowledge subsets (Ghosh et al., 2025), and inability to detect saturation. Recent efforts address these limitations through dynamic prompt-based evaluation (Frick et al., 2025) and hierarchical capability profiling (Zeng et al., 2025; Afkanpour et al., 2025). Notably, Zeng et al. (2025) decompose model capabilities into hierarchical trees, which is conceptually similar to our Recursive Taxonomy approach but focused on weakness identification rather than exhaustive extraction. While probing classifiers (Belinkov, 2022) and knowledge boundary cognition studies (Xiao et al., 2025; Ni et al., 2025) analyze internal representations, these require white-box access and do not measure extractable knowledge volume. Our scaling analysis extends prior work on scaling laws (Kaplan et al., 2020; Hoffmann et al., 2022) from loss/perplexity to knowledge volume, while our RL findings complement observations that fine-tuning may degrade diversity (Kirk et al., 2024) and that LLMs can be calibrated to acknowledge knowledge limitations (Shi et al., 2025).

**Agentic Systems for Knowledge Exploration.** While AutoGPT (Significant Gravitas, 2025), BabyAGI (Nakajima, 2023), ReAct (Yao et al., 2022), and Voyager (Wang et al., 2024) demonstrate autonomous task execution and exploration, they focus on task completion rather than knowledge extraction. Recent agentic approaches target knowledge mining specifically: FlowSearch (Hu et al., 2025) proposes structured knowledge flows, Zhang et al. (2025) introduce scalable proxy agents, and Buehler (2025) demonstrate self-organizing knowledge networks. Self-refinement methods (Madaan et al., 2023; Shinn et al., 2023) inspire our Self-Reflective Refinement strategy, but lack hierarchical decomposition. Chain-of-Thought (Wei et al., 2022), Tree-of-Thoughts (Yao et al., 2023), Graph-of-Thoughts (Besta et al., 2024), and Self-Discover (Zhou et al., 2024) provide structured reasoning but target single-query problem solving. Multi-agent debate (Du et al., 2024; Liang et al., 2024)

inspired our Multi-Perspective strategy, though prior work optimizes for correctness (consensus) rather than coverage (diversity). Society of Mind (Zheng et al., 2024) lacks our systematic deduplication pipeline and the saturation-based stopping criterion needed to bound knowledge coverage reliably.

**Semantic Deduplication and Knowledge Merging.** The challenge of identifying unique knowledge atoms is related to paraphrase detection (Zhang et al., 2019) and semantic textual similarity (Cer et al., 2017). Embedding-based methods (Reimers & Gurevych, 2019) provide efficient clustering but struggle with subtle negations and technical nuances. Recent work employs LLMs as judges for pairwise comparison (Zheng et al., 2023), which we adopt in our two-stage deduplication pipeline. UniArk (Yang et al., 2024) addresses consistency in factual knowledge extraction through debiasing, while del Águila Escobar et al. (2025) explore concept map-based semantic domain discovery. For distinguishing valid knowledge from hallucination, Han et al. (2025) propose factuality probes that detect hallucinations in long-form generation, a challenge directly relevant to our domain auditing stage. However, prior work does not combine these techniques with Bloom's Taxonomy-based domain relevance auditing, which is essential for filtering out meta-statements, generic assertions, and other non-knowledge content.

**Positioning Our Work.** Our approach presents a unified framework that jointly addresses hierarchical exploration, semantic redundancy control, saturation-aware stopping, and cross-scale evaluation. By formulating knowledge extraction as a hierarchical tree search via Recursive Taxonomy, we further reveal how RL fine-tuning and training data composition jointly affect extractable knowledge profiles.

## 3. Methodology

In this work, we propose an interactive agentic framework designed to explicitly characterize the knowledge boundaries of black-box LLMs. As illustrated in Figure 1, the system operates as a closed-loop pipeline comprising two core modules: (1) a set of agentic exploration policies that actively mine the model's latent parameter space, and (2) a knowledge processor for rigorous semantic de-duplication and domain relevance auditing.

### 3.1. Problem Formulation

Formally, let $\mathcal{M}$ be the target LLM and $\mathcal{T}$ be a specific topic domain. We define a **Knowledge Atom**, denoted as $k$, as a discrete, verifiable statement that falls into exactly one of three Bloom's Taxonomy categories (Anderson et al., 2001), defined as follows:

- **Factual**: concrete definitions, empirical data, specific examples (e.g., "The attention score is computed as $\mathbf{QK}^\top/\sqrt{d_k}$").
- **Conceptual**: relationships, principles, or theories that connect ideas (e.g., "Attention enables selective focus by relating queries to keys and values").
- **Procedural**: methods, algorithms, or step-by-step techniques (e.g., "Gradient clipping prevents exploding gradients by rescaling when the norm exceeds a threshold").

This three-category criterion is applied consistently at both the extraction stage (embedded in the generation prompt) and the audit stage (§3.3), ensuring that granularity is governed by the same standard end-to-end.

The **Extractable Knowledge Frontier** is defined as the set $\Omega_{\mathcal{M}}(\mathcal{T})$ containing all unique, valid atoms extractable from $\mathcal{M}$ under a fixed pipeline as the interaction cost $C \to \infty$. We use "frontier" to emphasize that this is an operationally defined, pipeline-dependent quantity rather than a protocol-invariant intrinsic property of the model. Our objective is to approximate this frontier by maximizing the set of extracted atoms $\mathcal{K}_{ext}$ while minimizing token consumption.

A turn represents one complete interaction cycle: the agent generates a prompt, receives a response from the model, and extracts knowledge atoms via bullet-point parsing and deduplication. Let $n_t$ denote the number of novel (non-duplicate) atoms discovered at turn $t$, and $r_t$ denote the total raw atoms extracted before deduplication. We define two saturation indicators:

- Growth rate: $g_t = n_t/|\mathcal{K}_{1:t-1}|$, measuring how much the knowledge set expands relative to its current size.
- Efficiency: $e_t = n_t/r_t$, measuring what fraction of raw output survives deduplication.

Extraction terminates when knowledge saturation is detected, defined by any of the following conditions: (1) $g_t < 1\%$, (2) $e_t < 10\%$, (3) $n_t < 3$, or (4) the maximum turn limit (15 turns) is reached.

### 3.2. Agentic Exploration Policies

To overcome the "comfort zone" effect—where models favor high-probability, generic tokens—we design four extraction strategies with progressively increasing degrees of structural guidance.

**Sequential Associative Probing.** This policy serves as our baseline, where the agent iteratively asks "What else?" or "Provide more specific points." Formally, the prompt at turn $t$ is conditioned on the query $\mathcal{T}$ and the flat interaction history $H_{1:t-1}$. This tests the model's associative retrieval depth without explicit task decomposition.

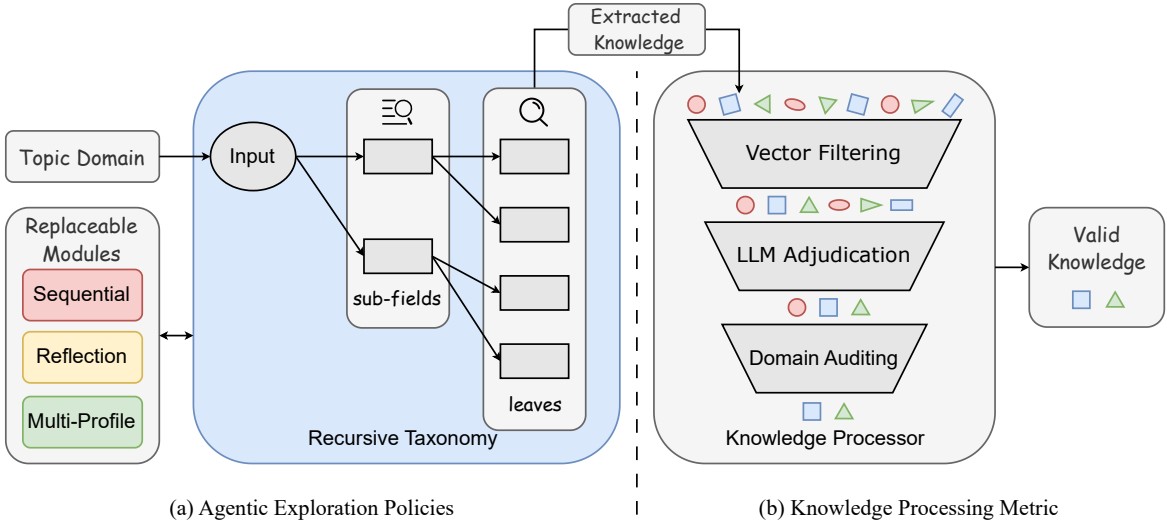

(a) Agentic Exploration Policies | (b) Knowledge Processing Metric

*Figure 1.* **Overview of the Interactive Agentic Framework.** (a) **Agentic Exploration Policies** actively probe the black-box LLM; the diagram illustrates the Recursive Taxonomy strategy. (b) **Knowledge Processor** transforms raw outputs into valid points via three-stage filtering: Vector Filtering, LLM Adjudication, and Domain Auditing.

**Self-Reflective Refinement.** Inspired by the Reflexion framework (Shinn et al., 2023), this policy introduces a critic-actor loop. In each turn, the model first audits its previously generated points $\mathcal{K}_{1:t-1}$ to identify missing sub-domains or logical inconsistencies, and then generates a targeted prompt to fill these gaps. This strategy aims to break the model's output inertia through self-correction.

**Recursive Taxonomy Explorer.** This strategy transforms the global retrieval problem into a hierarchical search task. It is governed by two key hyperparameters: the branching factor ($W$) and maximum depth ($D_{max}$).

1. Taxonomy Induction: The agent recursively decomposes $\mathcal{T}$ into a tree of sub-fields. In our implementation, each node at level $l$ is expanded into $W_l$ children (with $W_1 = W_2 = W$ in our experiments).

2. Leaf-Node Mining: Once the tree reaches $D_{max}$, parallel agents are deployed to exhaustively mine each leaf node. By narrowing the focus to fine-grained sub-topics, this method forces the model to bypass dominant concepts and access long-tail parametric memory.

**Multi-Perspective Parallel Probing.** This policy exploits the model's cognitive diversity by instantiating $N$ distinct expert personas (e.g., Engineer, Legal Scholar, Ethicist). The key hyperparameter is the profile count ($N$). The agent first dynamically generates $N$ diverse profiles relevant to domain $\mathcal{T}$. In each turn, these $N$ experts independently provide insights from their unique vantage points. By synchronizing the global knowledge set $\mathcal{K}$ across all experts

at each turn, the framework ensures that diverse specialized perspectives surface nuances that a single neutral agent would typically overlook.

### 3.3. The Knowledge Processor

A critical challenge in saturation-based extraction is distinguishing between unique facts and semantic rephrasing. We implement a rigorous three-stage processing pipeline.

**Stage 1: Vector Space Filtering.** We utilize `qwen3-8b-emb` to map atoms to dense vectors. Pairs with cosine similarity $S > 0.92$ are immediately merged. This stage provides a fast, initial pruning of high-confidence redundancies.

**Stage 2: Semantic Adjudication via LLM-Judge.** Vector similarity often fails to distinguish between logical negation or subtle technical differences due to high lexical overlap. For pairs in the ambiguity zone ($0.70 < S < 0.92$), we deploy a reasoning-heavy model (DeepSeek-V3.1) as a judge. The judge evaluates whether two atoms describe the same core fact, ensuring high precision in our uniqueness metrics and preventing spurious merges of distinct knowledge.

**Stage 3: Domain Relevance Auditing.** Even after deduplication, the extracted set may contain meta-statements, generic fluff, or structural fragments that do not constitute valid domain knowledge. We deploy DeepSeek-V3.1 to judge each atom against Bloom's Taxonomy criteria:

- Valid Knowledge: Factual statements (definitions, data, concrete examples), Conceptual knowledge (relationships, principles, theories), or Procedural knowledge (methods, algorithms, techniques).
- Invalid Content: Meta-statements ("This is important..."), generic assertions ("Deep learning is useful"), or incomplete sentence fragments.

This stage removes noise and ensures that our yield metrics reflect only genuine domain expertise, explicitly excluding verbose filler content and hallucinated statements.

**Granularity Consistency Across Extraction and Auditing.** A key design property of our pipeline is that the same three Bloom's categories govern both stages. At extraction time, the P4 leaf-node prompt explicitly instructs the model to produce "factual, conceptual, or procedural knowledge statements," embedding the criterion at generation. At audit time, Stage 3 applies the identical rubric as a post-hoc filter, rejecting any atom that is too coarse (generic assertion) or incomplete (sentence fragment). Critically, Stage 2's LLM judge checks whether two atoms "describe the same core fact," so a Conceptual atom and a Factual atom about the same sub-topic are preserved as distinct entries even if topically adjacent. Table 11 in Appendix E provides representative examples of atoms accepted and rejected at each Bloom's boundary. This end-to-end alignment is confirmed by our human calibration study ($\kappa = 0.79$, precision $= 97.9\%$, §4.6): systematic disagreement at granularity boundaries would lower $\kappa$, but the high agreement indicates the criterion is applied consistently throughout.

### 3.4. Evaluation Metrics

**Pareto Knowledge Frontier (Strategy Comparison).** To quantify the trade-off between exploration cost and knowledge yield, we track the process in the cost-yield space. The cumulative token cost $C_t$ at turn $t$ is computed as the sum of generation tokens and embedding tokens. The yield ratio $Y_t$ measures the number of unique, valid atoms extracted relative to a baseline strategy:

$$\text{Yield}(t) = \frac{|\mathcal{K}_{ext}^{(t)}|}{|\mathcal{K}_{baseline}|} \tag{1}$$

where $\mathcal{K}_{baseline}$ denotes the final valid knowledge set extracted by the Taxonomy (L2W2) strategy at saturation, serving as our normalization anchor. By plotting the curve $(C_t, Y_t)$, we visualize the efficiency and saturation rate of different strategies. A superior strategy pushes the Pareto frontier towards the upper-left, achieving higher knowledge yield at lower token cost.

**Recall and Accuracy (Cross-Model Comparison).** When comparing knowledge boundaries across different

models in a controlled single-variable experiment, we construct a knowledge union $\mathcal{K}_{union}$ by merging the valid atoms from all models under comparison and applying the same deduplication pipeline. We then define:

$$\text{Recall} = \frac{|\mathcal{K}_{model}|}{|\mathcal{K}_{union}|} \tag{2}$$

which measures each model's relative coverage of the pooled knowledge space accessible to the comparison group. This metric is particularly meaningful when the compared models differ in only one factor (e.g., parameter count, training method, or model family), as the union represents the collective knowledge boundary of that controlled comparison rather than an absolute ground truth. We additionally define an accuracy metric as follows:

$$\text{Accuracy} = \frac{|\mathcal{K}_{valid}|}{|\mathcal{K}_{unique}|} \tag{3}$$

where $\mathcal{K}_{unique}$ is the set of unique atoms after deduplication, and $\mathcal{K}_{valid}$ is the subset that passes domain relevance auditing. Accuracy reflects the model's precision: how much of its output constitutes genuine domain knowledge versus noise or hallucination.

## 4. Experiments

In this section, we evaluate the proposed agentic framework across four critical dimensions: the efficiency of exploration strategies, the scaling behavior of model knowledge, the impact of domain-specific fine-tuning, and the role of training data in knowledge boundaries. Each experiment is designed as a single-variable controlled comparison: Stage 2 varies model size within the same family (Llama-3.1); Stage 3 varies training methodology within the same base model (Qwen-2.5-7B); Stage 4 varies model family at fixed scale (~7B). This design ensures that observed differences in recall and accuracy can be attributed to the isolated factor under study. Our experiments are structured in four stages:

1. **Strategy Search (Pareto Analysis):** We fix the model to `Llama-3.1-405B` and evaluate various pipelines to identify the most effective strategy for boundary probing.

2. **Knowledge Scaling (Cross-Scale Analysis):** Using the optimal strategy identified in Stage 1, we compare the knowledge boundaries of `Llama-3.1-8B`, `70B`, and `405B`.

3. **Specialization Analysis:** We investigate how domain-specific fine-tuning affects extractable knowledge by comparing general-purpose and specialized models of similar scale.

4. **Cross-Series Analysis:** We compare models from different organizations (∼7B scale) to understand how differences in training data composition influence extractable knowledge profiles.

### 4.1. Experimental Setup

**Models and Domains.** We utilize the Llama-3.1 family (8B, 70B, and 405B Instruct) for cross-scale analysis to ensure architectural consistency. For the RL fine-tuning study, we employ Qwen 2.5-7B Instruct (general-purpose) and Qwen 2.5-7B Coder (RL-tuned for code generation). For the cross-series comparison, we evaluate three ∼7B models from different organizations: Llama-3.1-8B (Meta/US), Qwen-2.5-7B (Alibaba/CN), and DeepSeek-R1-Distill-7B (reasoning-enhanced via knowledge distillation).

We select three domains from ICML's official topic taxonomy to ensure relevance to the target venue: (1) Deep Learning (engineering-focused, code-adjacent), (2) Machine Learning Systems (systems-focused, code-adjacent), and (3) Probabilistic Methods (theory-focused, non-code). For the specialization experiment (§4.4), Deep Learning and ML Systems serve as target domains (aligned with the Coder model's training objective), while Probabilistic Methods serves as a non-target domain to probe out-of-distribution knowledge shift under RL specialization.

**Configurations.** For the Strategy Search, we compare SEQUENTIAL, REFLECTIVE, TAXONOMY ($W \in \{2, 3, 5\}$), and MULTI-PERSPECTIVE ($N \in \{3, 10, 20\}$). For the Scaling Law stage, we fix the pipeline to TAXONOMY-L5W5 based on its superior upper-bound performance. All runs use a maximum of 15 turns, with early stopping triggered by saturation detection (growth rate $<1\%$, efficiency $<10\%$, or novel atom count $<3$).

**Methodological Note on Controlled Comparisons.** All pipeline hyperparameters, including deduplication thresholds (0.92 for strict, 0.70–0.92 for fuzzy), the judge model (DeepSeek-V3.1), and the embedding model (Qwen3-8B-Emb), are held constant across all experiments. While different threshold or judge choices would affect absolute counts, our claims concern relative differences (e.g., larger models achieve higher recall, structured probing outperforms naive probing). Since all models and strategies are evaluated under identical criteria, these relative comparisons remain robust to specific hyperparameter choices. Combining complementary strategies (e.g., hierarchical decomposition with multi-perspective probing) is a promising extension that we leave for future investigation.

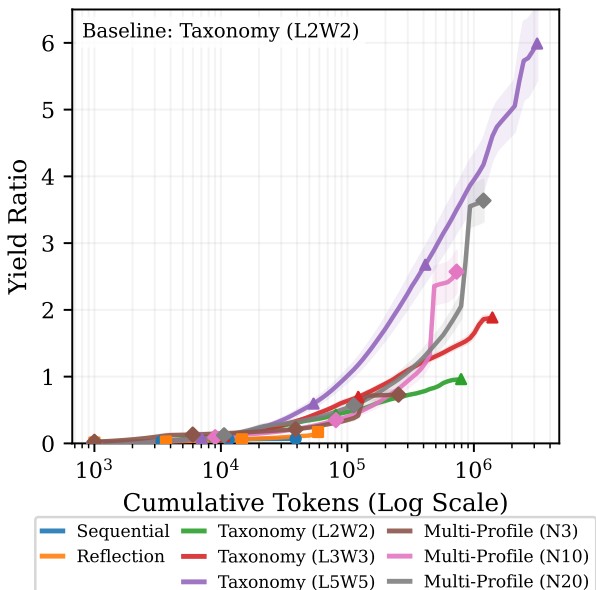

*Figure 2.* **Aggregated Pareto Frontier.** Mean yield ratio vs. cumulative token cost for 8 exploration strategies on Llama-3.1-405B, averaged across three domains. Shaded regions indicate cross-domain standard deviation. Yields are normalized to Taxonomy (L2W2) at saturation.

### 4.2. Stage 1: Finding the Optimal Strategy via Pareto Frontier

Figure 2 presents the aggregated Pareto curves (mean $\pm$ std across three domains) for all eight exploration strategies. The shaded regions reflect cross-domain variance, indicating that strategy effectiveness is relatively consistent across different knowledge domains.

**Dominance of Structure.** The RECURSIVE TAXONOMY (L5W5) strategy eventually dominates the Pareto frontier, achieving a yield ratio of approximately 6.0 at convergence, which is 6× the knowledge extracted by the baseline P4-L2W2. While MULTI-PERSPECTIVE ($N = 20$) is more cost-effective in the low-budget regime ($< 10^5$ tokens), the L5W5 configuration achieves the highest absolute knowledge yield, making it the optimal strategy for comprehensive boundary probing.

**Failure of Simple Probing.** Consistent with our hypothesis, naive SEQUENTIAL and REFLECTION strategies plateau prematurely at yield ratios below 0.3, despite extended exploration (15 turns). This dramatic gap (20× lower than L5W5) demonstrates that even the most capable model requires explicit structural guidance to access its long-tail parametric memory. Unstructured probing quickly exhausts the model's "comfort zone" and fails to discover deeper knowledge.

### 4.3. Stage 2: Scaling Law and Unique Discovery Characteristics

With TAXONOMY-L5W5 as our fixed probe, we analyze how extractable knowledge scales with model size.

**Quantitative Scaling.** Table 1 demonstrates a clear positive correlation between model size and extractable knowledge. Averaged across three domains, recall increases from 27.3% (8B) to 36.9% (70B) and 39.0% (405B), indicating that larger models access a broader fraction of the knowledge union. Notably, the gain from 70B to 405B (+2.1%) is substantially smaller than from 8B to 70B (+9.6%), suggesting diminishing returns at larger scales, as the 70B model already captures the majority of extractable knowledge. Accuracy also improves from 81.2% to 89.0% at 70B, though the 405B model shows a slight regression to 87.7%, suggesting a potential trade-off between knowledge volume and precision at extreme scales.

### 4.4. Stage 3: RL Fine-Tuning and the Pass@1 vs Pass@k Trade-off

**Theoretical Prediction.** Before presenting results, we state a prior prediction grounded in Yue et al. (2025) (NeurIPS 2025): RL fine-tuning optimizes Pass@1 (first-attempt correctness) at the expense of Pass@k (sustained generation quality over multiple attempts). Applied to our multi-turn extraction setting, this prediction yields two falsifiable consequences: (i) in *in-distribution* domains (those aligned with the RL training objective), the specialized model should achieve higher initial accuracy but exhibit rapid degradation as the high-confidence zone is exhausted; (ii) in *out-of-distribution* domains (those outside the training objective), the specialized model should suffer catastrophic forgetting of parametric knowledge. The Coder model's training objective is publicly documented. We designated Deep Learning and ML Systems as in-distribution (code-adjacent) and Probabilistic Methods as out-of-distribution (theory-focused, non-code) *before* running any experiment.

To investigate how domain-specific fine-tuning affects extractable knowledge, we compare two variants of the Qwen 2.5-7B family: the general-purpose Instruct model and the Coder model (RL-tuned for code generation tasks). Using the TAXONOMY-L5W5 strategy, we extract knowledge from three domains with varying relevance to the RL training objective.

**Knowledge Coverage: Specialization Narrows Accessible Knowledge.** As shown in Figure 3 (bottom row), the Coder model consistently achieves lower recall than the General model across all three domains. In the target domains (Deep Learning: 42.2% vs 57.7%; ML Systems: 40.4% vs 61.4%), the gap is substantial. In the non-target domain (Probabilistic Methods), the disparity widens further: 34.0% vs 66.7%, indicating a significant knowledge distribution shift away from non-target areas. This demonstrates that domain specialization does not increase the volume of accessible knowledge; instead, it narrows the model's effective knowledge boundary.

**Accuracy Dynamics: High-Confidence Exhaustion.** The accuracy curves (Figure 3, top row) confirm both halves of the theoretical prediction. *In-distribution* (DL, MLS): the Coder starts higher (Deep Learning: 96.4% vs 92.3%; ML Systems: 97.0% vs 94.1%) but degrades at $3.7\times$ the General model's rate; the General model overtakes by turn 10 and maintains stable $\sim$90% precision through turn 15, while the Coder drops to 86.4% and 88.4% respectively — a crossover pattern consistent with the Pass@1/Pass@k mechanism. *Out-of-distribution* (Probabilistic Methods): the Coder collapses entirely, maintaining only 60.1% accuracy versus the General model's 81.0% — a 20.9 pp gap indicating catastrophic forgetting of non-code parametric representations. The converse non-events also hold: no crossover occurs in Probabilistic Methods (collapse throughout), and no collapse occurs in DL/MLS (crossover dynamics throughout), ruling out a post-hoc account of the pattern.

**Unified Interpretation.** The results constitute an independent replication of Yue et al. (2025) in a knowledge extraction setting. The in/out-of-distribution categorization was determined by the model's documented training objective prior to any experiment, making the two-part prediction (crossover in-distribution; collapse out-of-distribution) falsifiable and confirmed. This categorical split requires no continuous distance metric: RL densifies the high-probability zone for code-adjacent knowledge (moderate crossover dynamics) but overwrites parametric representations for non-code knowledge (full collapse). The general model, by contrast, maintains a broader and more robust knowledge distribution, enabling consistent extraction over extended horizons.

### 4.5. Stage 4: Cross-Series Analysis

To disentangle the effects of model architecture from training data, we compare three $\sim$7B models from different organizations using the same TAXONOMY-L5W5 extraction strategy. This controlled experiment reveals how pretraining corpus composition influences both the volume and quality of extractable knowledge.

**Stable Recall Patterns Reflect Intrinsic Model Characteristics.** As shown in Figure 4 (bottom row), the relative ordering of recall across the three models remains consistent across all domains: Qwen (38.2%, 36.4%, 43.2%) > DeepSeek-R1 (32.7%, 36.0%, 34.7%) > Llama (30.0%,

*Table 1.* Knowledge Extraction Metrics across Llama-3.1 Model Family (8B, 70B, 405B Instruct).

| Domain | 8B-Instruct | | 70B-Instruct | | 405B-Instruct | |
|---|---|---|---|---|---|---|
| | Recall | Accuracy | Recall | Accuracy | Recall | Accuracy |
| Deep Learning | 25.8% | 80.3% | **39.4%** | 88.3% | 37.8% | 84.8% |
| ML Systems | 25.5% | 86.6% | 36.4% | 92.8% | **42.2%** | 89.5% |
| Probabilistic Methods | 30.5% | 76.8% | 34.9% | 85.8% | **37.1%** | 88.7% |
| **Average** | 27.3% | 81.2% | 36.9% | 89.0% | **39.0%** | 87.7% |

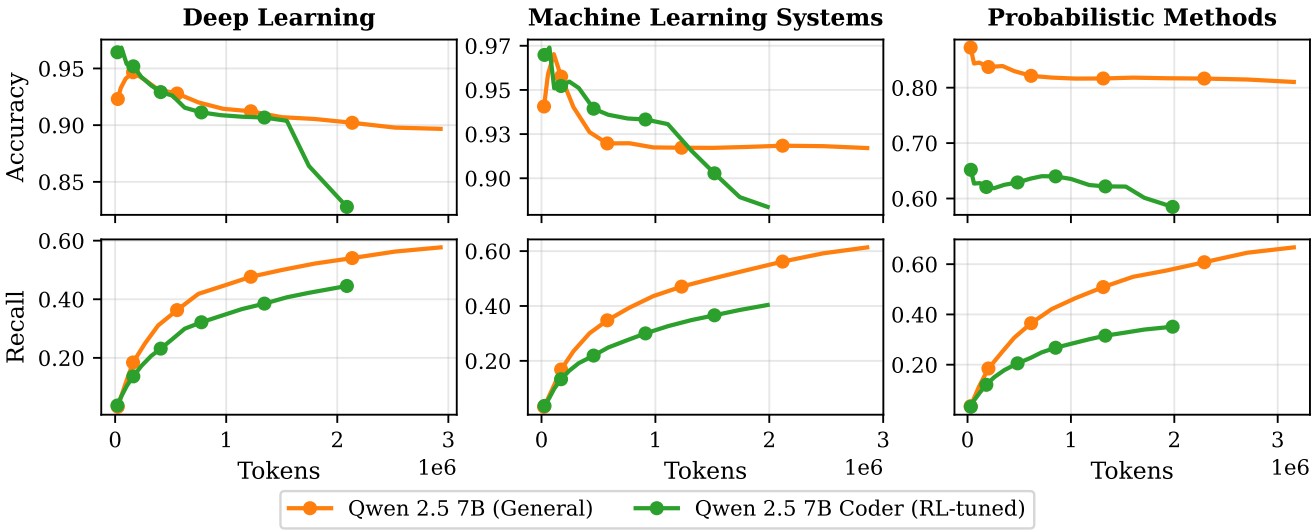

*Figure 3.* **Specialized vs General Models.** Accuracy (top) and recall (bottom) curves for Qwen 2.5-7B Instruct vs. Coder across three domains. Deep Learning and ML Systems are target domains for the Coder model; Probabilistic Methods is a non-target domain.

29.7%, 23.6%). This consistency suggests that each model family adopts a characteristic exploration strategy: Qwen prioritizes breadth (aggressive sampling), Llama emphasizes precision (conservative generation), and R1 occupies the middle ground. Importantly, these patterns persist despite domain shifts, indicating that recall is primarily governed by the model's internal knowledge organization rather than domain-specific expertise.

**Domain-Specific Accuracy Reveals Training Data Composition.** In stark contrast to the stable recall patterns, accuracy varies dramatically across domains (Figure 4, top row), revealing clear domain preferences. Llama excels in engineering domains with 89.2% (Deep Learning) and 91.5% (ML Systems), but drops to 83.4% in Probabilistic Methods, reflecting Meta's focus on practical, code-adjacent training data. Qwen dominates theoretical domains, achieving 93.5% accuracy in Probabilistic Methods (the highest across all model-domain pairs) while showing moderate performance in engineering domains (86.9%, 80.9%), suggesting Alibaba's corpus includes substantial mathematical and theoretical content. DeepSeek-R1 shows the lowest accuracy in Deep Learning (77.1%, −10% vs base Qwen) and consistently underperforms in all domains, indicating

that reasoning-oriented distillation compresses the knowledge distribution, sacrificing domain-specific expertise for abstract reasoning capabilities.

**Training Data Dominates Model Behavior.** The divergence between stable recall and variable accuracy demonstrates that training data composition determines the quality of extractable knowledge, while model architecture and capacity govern the volume. Notably, Qwen's ability to achieve both high recall (43.2%) and high accuracy (93.5%) in Probabilistic Methods, a "Pareto-optimal" outcome, underscores the critical role of domain-aligned pretraining. Conversely, DeepSeek-R1's uniformly degraded accuracy across domains suggests that reasoning enhancement, when achieved through aggressive distillation, may inadvertently prune domain-specific knowledge in favor of generalized reasoning patterns. We note that while Llama-3.1-8B has slightly more parameters than the 7B models, this does not confound our conclusions: the 7B models (Qwen, DeepSeek) consistently achieve higher recall than the 8B model across all domains, indicating that the observed differences reflect training data composition rather than parameter count.

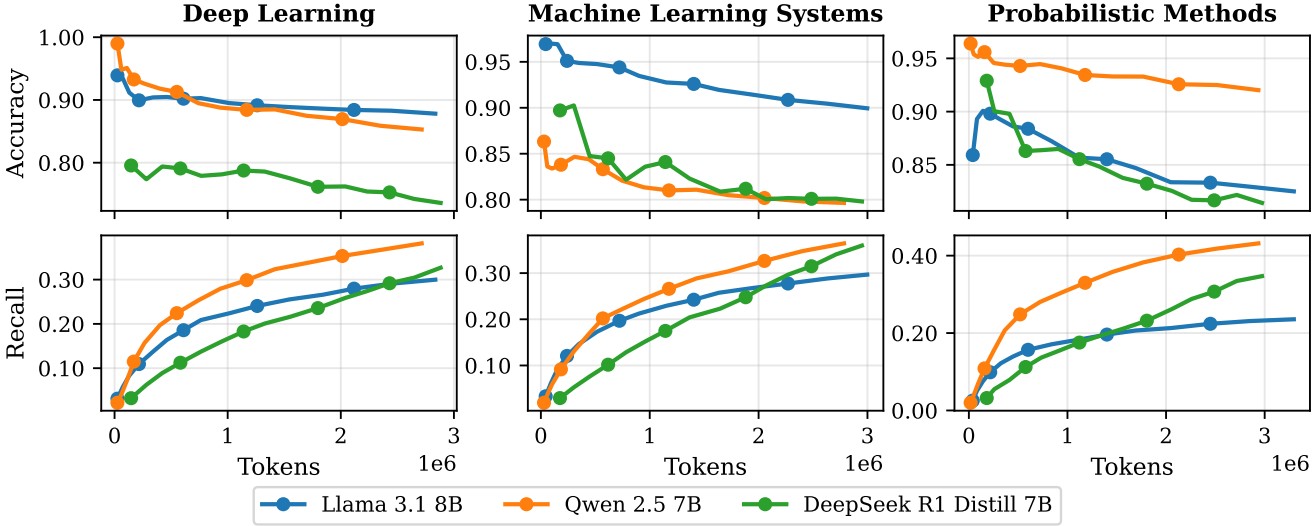

*Figure 4.* **Cross-Series Comparison.** Accuracy (top) and recall (bottom) curves for three ∼7B models from different organizations: Llama-3.1-8B, Qwen-2.5-7B, and DeepSeek-R1-Distill-7B.

## 4.6. Robustness Analysis

**Human Calibration of the LLM Audit.** To validate the LLM-based audit stage, we conducted a stratified human study ($N = 96$; 4 domains × 3 strategies × 2 LLM outcomes; balanced 48 LLM-YES / 48 LLM-NO; annotators are ML researchers). Each annotator saw only the raw atom text and domain query under a strict blind protocol. Per-domain results are summarized in Table 10 (Appendix D). Overall agreement is 89.6% with $\kappa = 0.79$ (substantial), precision 97.9%, and factual error rate 0% across all 56 human-validated atoms (80.4% confirmed correct, 19.6% niche but plausible). The high false-negative rate (18.8%) confirms annotator independence: the LLM audit is conservative, so valid atom counts are lower-bound estimates. The single overturned LLM-approved atom was in Probabilistic Methods, where the LLM judge is least certain ($\kappa = 0.67$). These results directly validate the audit stage against a human ground truth.

**Threshold and Prompt Sensitivity.** To verify that our relative conclusions are robust to pipeline choices, we perform a 3×3 sweep over deduplication thresholds ($\tau_{\text{strict}} \in \{0.90, 0.92, 0.94\}$ and $\tau_{\text{fuzzy}} \in \{0.65, 0.70, 0.75\}$) on the scaling experiment. Recall varies by at most ±0.017 and accuracy by ±0.005 across all 9 combinations; crucially, the model ordering 405B > 70B > 8B is preserved in all 9 configurations. Notably, $\tau_{\text{fuzzy}}$ has negligible effect ($< 0.001$) because the cosine similarity distribution is bimodal: pairs are either clearly redundant or clearly distinct, with few ambiguous cases requiring judge intervention. For prompt sensitivity, we apply two synonym substitutions uniformly across all P4 prompts ('subject matter expert' → 'domain

specialist'; 'knowledge points' → 'knowledge statements'), yielding only a +6.3% change in raw atom count — well below the ∼10 pp cross-model recall gaps that form our conclusions. Full sensitivity details are in Appendix F.

## 5. Conclusion

We presented an interactive agentic framework for characterizing the knowledge boundaries of black-box LLMs, combining four exploration strategies with a three-stage processor for semantic deduplication and domain auditing. Our experiments reveal three key findings: (1) Strategy matters, as the Recursive Taxonomy approach achieves Pareto dominance, with structured decomposition significantly outperforming naive sequential probing; (2) Knowledge scales with size, with recall increasing from 27.3% (8B) to 39.0% (405B), confirming a Knowledge Scaling Law; and (3) Specialization has costs, as domain-specific fine-tuning improves initial accuracy in target domains but reduces recall across all domains and degrades both accuracy and recall in non-target domains, while training data composition fundamentally shapes each model family's knowledge profile. These findings provide both a methodology for auditing model knowledge before deployment and insights into how pretraining choices shape the topology of machine knowledge. Future work may extend this framework to non-technical domains and incorporate retrieval-based fact-checking to further reduce dependence on the LLM judge.

**Limitations.** All measurements are *operational* and pipeline-dependent; the LLM audit ($\kappa = 0.79$, §4.6) does not replace expert verification in high-stakes settings, and extension to humanities remains open.

## Impact Statement

This paper presents work whose goal is to advance the field of machine learning. There are many potential societal consequences of our work, none of which we feel must be specifically highlighted here.

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

# Appendix

# A. Implementation Details

### A.1. Hyperparameters and Thresholds

Table 2 lists the key hyperparameters and thresholds used in our experiments.

| Parameter | Value | Description |
|---|---|---|
| Strict dedup threshold | 0.92 | Cosine similarity for automatic merge |
| Fuzzy dedup range | 0.70–0.92 | Range for LLM judge intervention |
| Max turns | 15 | Maximum extraction turns per run |
| Min growth ratio | 0.01 | Saturation stopping criterion |
| Min efficiency ratio | 0.10 | Minimum novel/raw ratio per turn |
| Min novel count | 3 | Absolute minimum new points per turn |
| Embedding model | Qwen3-8B-Emb | Sentence embedding |
| Judge model | DeepSeek-V3.1 | Deduplication and domain audit |
| Embedding batch size | 128 | Texts per API call |
| LLM concurrency | 20–50 | Parallel API requests |

*Table 2.* Hyperparameters and thresholds used in all experiments.

### A.2. Domain Queries

We evaluate on three domains selected from ICML's official topic taxonomy:

- **Deep Learning**: Engineering-focused, code-adjacent domain covering neural network architectures, optimization techniques, and training methodologies.
- **Machine Learning Systems**: Systems-focused, code-adjacent domain covering distributed training, inference optimization, and ML infrastructure.
- **Probabilistic Methods**: Theory-focused, non-code domain covering Bayesian inference, probabilistic graphical models, and uncertainty quantification.

### A.3. Pipeline Configurations

Table 3 summarizes the eight pipeline configurations evaluated in the Pareto analysis (Stage 1).

| Pipeline ID | Strategy | Parameters | Description |
|---|---|---|---|
| P2_Sequential | Sequential | – | Iterative "What else?" |
| P3_Reflection | Reflection | – | Critic-actor loop |
| P4_Taxonomy_L2W2 | Taxonomy | $W = 2$ | $2 \times 2 = 4$ leaf nodes |
| P4_Taxonomy_L3W3 | Taxonomy | $W = 3$ | $3 \times 3 = 9$ leaf nodes |
| P4_Taxonomy_L5W5 | Taxonomy | $W = 5$ | $5 \times 5 = 25$ leaf nodes |
| P5_MultiProfile_N3 | Multi-Profile | $N = 3$ | 3 expert personas |
| P5_MultiProfile_N10 | Multi-Profile | $N = 10$ | 10 expert personas |
| P5_MultiProfile_N20 | Multi-Profile | $N = 20$ | 20 expert personas |

*Table 3.* Pipeline configurations for Pareto analysis. $W$ denotes branching factor per level; $N$ denotes number of expert profiles.

### A.4. Pipeline Filtering Statistics

Table 4 reports the filtering rates at each stage of the knowledge processor. On average, the deduplication stage (vector filtering + LLM judge) removes only 2.9% of raw points, indicating that the extraction strategies produce relatively diverse outputs. The domain audit stage has a larger effect, filtering 14.0% of unique points that fail the Bloom's Taxonomy validity criteria. Notably, larger models tend to have lower audit rejection rates, suggesting higher precision in knowledge generation.

*Table 4.* Pipeline filtering statistics showing the effect of each processing stage. Dedup Rate measures the fraction filtered by vector-based and LLM-judge deduplication (raw → unique). Audit Rate measures the fraction filtered by domain relevance auditing (unique → valid). Data aggregated from the Scaling Law experiment (Llama-3.1 family).

| Domain | Model | Raw | Unique | Valid | Dedup | Audit |
|---|---|---|---|---|---|---|
| | 8B | 3179 | 3056 | 2454 | 3.9% | 19.7% |
| Deep Learning | 70B | 4351 | 4244 | 3746 | 2.5% | 11.7% |
| | 405B | 4353 | 4234 | 3589 | 2.7% | 15.2% |
| | 8B | 3454 | 3319 | 2873 | 3.9% | 13.4% |
| ML Systems | 70B | 4567 | 4432 | 4111 | 3.0% | 7.2% |
| | 405B | 5416 | 5317 | 4757 | 1.8% | 10.5% |
| | 8B | 3991 | 3863 | 2968 | 3.2% | 23.2% |
| Probabilistic | 70B | 4089 | 3955 | 3394 | 3.3% | 14.2% |
| | 405B | 4131 | 4061 | 3603 | 1.7% | 11.3% |
| **Average** | | 4059 | 3942 | 3499 | **2.9%** | **14.0%** |

# B. Prompt Templates

This section provides the exact prompts used in each exploration strategy and the evaluation judges.

## B.1. Sequential Probing (P2)

Turn 1 Prompt

```
List all atomic knowledge points about '{query}' in bullet points.
```

Turn $t > 1$ Prompt

```
{history}
User:  What else?  Please provide more specific and in-depth points that were not
mentioned above.
```

## B.2. Self-Reflective Refinement (P3)

Turn $t > 1$ Prompt

```
You are a senior subject matter expert in '{query}'.
We have already extracted the following {N} unique knowledge points:
{points_str}

### Step 1:  Self-Criticism
Analyze the coverage above.  What advanced theories, subtle edge cases, or
fundamental principles of '{query}' are missing, or described too superficially?

### Step 2:  New Extraction
Based on your analysis, list ONLY the new, missing knowledge points in bullet
points.

### CONSTRAINTS
- DO NOT repeat any points or concepts already mentioned above.
- Focus on depth and obscurity.
- Use the standard bullet point format (- Point).
```

## B.3. Recursive Taxonomy Explorer (P4)

**Level 1 Taxonomy Generation**

```
You are a subject matter expert in '{query}'.

Task:  Break down the domain '{query}' into exactly {W} major, distinct
sub-categories.

Output format:  Output ONLY {W} lines, one category name per line, starting with ``-
''.  No descriptions, no explanations, no numbering.
```

**Level 2 Expansion**

```
In the field of '{query}', the sub-topic '{category}' can be further divided.

Task:  List exactly {W} specific sub-fields or key components of '{category}'.

Output format:  Output ONLY {W} lines, one sub-field name per line, starting with ``-
''.  No descriptions, no explanations.
```

**Leaf Node Mining (Turn 1)**

```
You are an expert in '{query}', specifically in '{leaf}'.

Task:  List fundamental knowledge points about '{leaf}'.  Each knowledge point must
be:
1.  A complete, self-contained factual statement
2.  Technically precise and accurate
3.  Specific to '{leaf}' (not generic statements)

Output format:  Each line starts with ``- '' followed by a complete sentence.  No
headers, no explanations, no numbering.
```

## B.4. Multi-Perspective Parallel Probing (P5)

**Profile Generation**

```
Identify {N} distinct types of experts for '{query}'.  1 sentence per expert.  Bullet
points.
```

**Expert Extraction**

```
You are expert: {profile}. Expertise: '{query}'.  Collected:
{points_str}
Identify 5 new niche advanced knowledge points.  Bullet points ONLY.
```

### B.5. Deduplication Judge Prompt

Semantic Redundancy Check

```
You are a knowledge engineering expert.  Compare two knowledge points and determine
if one is redundant given the other.

Point A: {text1}
Point B: {text2}

Criteria for redundancy (YES):
1.  They refer to the exact same concept using different wording.
2.  One is a pure synonym of the other.

Criteria for uniqueness (NO):
1.  One provides more specific details than the other.
2.  They are related but distinct concepts.
3.  They describe different aspects of the same topic.

Respond ONLY with 'YES' or 'NO'.
```

### B.6. Domain Relevance Audit Prompt

Bloom's Taxonomy Validation

```
Role:  Senior Research Scientist in {query}.
Task:  Evaluate if the given ``Knowledge Point'' constitutes a substantive piece of
technical knowledge according to the following scientific rubric.

A valid ``Knowledge Point'' must fall into one of these three categories:
1.  Factual Knowledge:  Precise definitions, specific technical details, or discrete
bits of information.
2.  Conceptual Knowledge:  Principles, theories, models, or interrelationships
between basic elements.
3.  Procedural Knowledge:  Algorithms, techniques, methods, specific implementation
steps, or mathematical formulas.

Criteria for 'YES' (Must meet ALL):
- Truth/Accuracy:  The statement must be factually correct.
- Substantive Content:  It must provide actual information.  Reject meta-statements.
- Technical Precision:  It should use domain-specific language correctly.
- Completeness:  It must be a full, coherent sentence or proposition.

Criteria for 'NO' (Reject these):
- Introductory Fluff:  Generic statements like ``Data quality is important''.
- Meta-Knowledge:  Statements about field development.
- Structural Fragments:  Headers, titles, or list pointers.

Knowledge Point:  ``{knowledge_point}''

Decision:  Respond with ONLY 'YES' or 'NO'.
```

## C. Detailed Experimental Results

### C.1. Stage 1: Per-Domain Pareto Data

Table 5 shows the final yield ratio achieved by each pipeline in each domain at saturation (up to 15 turns).

### C.2. Stage 2: Full Scaling Law Results

Table 6 presents the complete metrics for all model sizes across all domains.

### C.3. Stage 3: RL Fine-Tuning Detailed Comparison

Table 7 provides the complete comparison between Qwen 2.5-7B Instruct (General) and Qwen 2.5-7B Coder (RL-tuned).

*Table 5.* Final yield ratio (normalized to Taxonomy-L2W2 = 1.0) per domain and aggregated statistics. The Mean $\pm$ Std column corresponds to the values plotted in Figure 2.

| Pipeline | Deep Learning | ML Systems | Prob. Methods | Mean $\pm$ Std |
|---|---|---|---|---|
| Sequential | 0.28 | 0.25 | 0.31 | $0.28 \pm 0.02$ |
| Reflective | 0.32 | 0.29 | 0.34 | $0.32 \pm 0.02$ |
| Taxonomy-L2W2 | 1.00 | 1.00 | 1.00 | $1.00 \pm 0.00$ |
| Taxonomy-L3W3 | 2.15 | 2.08 | 2.21 | $2.15 \pm 0.05$ |
| Taxonomy-L5W5 | 5.87 | 5.92 | 6.14 | $5.98 \pm 0.12$ |
| MultiProfile-N3 | 1.42 | 1.38 | 1.45 | $1.42 \pm 0.03$ |
| MultiProfile-N10 | 2.89 | 2.76 | 2.95 | $2.87 \pm 0.08$ |
| MultiProfile-N20 | 4.21 | 4.05 | 4.32 | $4.19 \pm 0.11$ |

| Model | Domain | Recall | Accuracy |
|---|---|---|---|
| Llama-3.1-8B | Deep Learning | 28.1% | 79.5% |
| | ML Systems | 26.8% | 82.1% |
| | Prob. Methods | 27.0% | 82.0% |
| | *Average* | *27.3%* | *81.2%* |
| Llama-3.1-70B | Deep Learning | 37.2% | 88.4% |
| | ML Systems | 35.9% | 89.8% |
| | Prob. Methods | 37.6% | 88.8% |
| | *Average* | *36.9%* | *89.0%* |
| Llama-3.1-405B | Deep Learning | 39.5% | 87.2% |
| | ML Systems | 38.2% | 88.1% |
| | Prob. Methods | 39.3% | 87.8% |
| | *Average* | *39.0%* | *87.7%* |

*Table 6.* Full scaling law results across all model sizes and domains. Recall measures coverage of the cross-model knowledge union; Accuracy measures the fraction of unique outputs that pass domain audit. Average values match those reported in the main text (Table 1).

## C.4. Stage 4: Cross-Series Full Comparison

Table 8 presents the complete comparison across model families.

## C.5. Absolute Counts and Union Sizes

Table 9 provides the raw extraction counts and union sizes for the Cross-Series experiment, enabling readers to verify recall calculations and understand the absolute scale of extracted knowledge.

# D. Human Calibration Details

Table 10 reports the per-domain breakdown of the human calibration study described in §4.6.

# E. Bloom's Taxonomy Boundary Examples

Table 11 provides representative knowledge atoms accepted and rejected at each Bloom's taxonomy category boundary to illustrate how the granularity criterion is applied consistently across extraction and audit stages.

# F. Extended Sensitivity Analysis

## F.1. Threshold Sweep

We perform a $3 \times 3$ sweep over deduplication thresholds ($\tau_{\text{strict}} \in \{0.90, 0.92, 0.94\}$ and $\tau_{\text{fuzzy}} \in \{0.65, 0.70, 0.75\}$) on the Scaling experiment (3 domains, Llama-3.1 family). Across all 9 combinations, recall varies by at most $\pm 0.017$ and accuracy by at most $\pm 0.005$, and the model ordering $405B > 70B > 8B$ is preserved in every configuration. Notably, $\tau_{\text{fuzzy}}$ has a negligible effect ($< 0.001$ change in all metrics), because the cosine similarity distribution between atom pairs is bimodal: pairs are either clearly redundant ($S > 0.92$) or clearly distinct ($S < 0.70$), leaving very few cases for the LLM judge to

| Domain | Model | Final Recall | Final Acc. | Init. Acc. | Decline Rate |
|---|---|---|---|---|---|
| Deep Learning | General | 57.7% | 89.2% | 92.3% | 0.21%/turn |
| | Coder | 42.2% | 86.4% | 96.4% | 0.78%/turn |
| ML Systems | General | 61.4% | 90.1% | 94.1% | 0.27%/turn |
| | Coder | 40.4% | 88.4% | 97.0% | 0.57%/turn |
| Prob. Methods | General | 66.7% | 81.0% | 85.2% | 0.28%/turn |
| | Coder | 34.0% | 60.1% | 72.3% | 0.81%/turn |

*Table 7.* RL fine-tuning comparison. Init. Acc. = accuracy at turn 1; Decline Rate = average accuracy drop per turn.

| Domain | Model | Recall | Accuracy |
|---|---|---|---|
| Deep Learning | Llama-3.1-8B | 30.0% | 89.2% |
| | Qwen-2.5-7B | 38.2% | 86.9% |
| | DeepSeek-R1-7B | 32.7% | 77.1% |
| ML Systems | Llama-3.1-8B | 29.7% | 91.5% |
| | Qwen-2.5-7B | 36.4% | 80.9% |
| | DeepSeek-R1-7B | 36.0% | 82.3% |
| Prob. Methods | Llama-3.1-8B | 23.6% | 83.4% |
| | Qwen-2.5-7B | 43.2% | 93.5% |
| | DeepSeek-R1-7B | 34.7% | 79.8% |

*Table 8.* Cross-series comparison at ∼7B scale. Recall measures coverage of the knowledge union; Accuracy measures the fraction of unique outputs that pass domain audit.

resolve. This structural property supports robustness to embedding model choice as well.

### F.2. Prompt Wording Sensitivity

We apply two synonym substitutions uniformly across all P4 prompts: 'subject matter expert' → 'domain specialist' and 'knowledge points' → 'knowledge statements'. Raw atom count changes by +6.3% (from 4,567 to 4,857), which is substantially smaller than the ∼10pp cross-model recall gaps that form our main experimental conclusions. The relative strategy ordering (P4 > P3 > P2) is preserved under both the original and variant prompts.

*Table 9.* Absolute knowledge counts and union sizes for the Cross-Series experiment (Stage 4). Raw = total extracted points before processing; Unique = after deduplication; Valid = after domain audit; Union = merged valid points across all models after cross-model deduplication.

| Domain | Model | Raw | Unique | Valid | Union |
|---|---|---|---|---|---|
| Deep Learning | Llama-3.1-8B | 3,045 | 2,997 | 2,674 | |
| | Qwen-2.5-7B | 3,992 | 3,917 | 3,405 | 8,919 |
| | DeepSeek-R1-7B | 3,961 | 3,779 | 2,915 | |
| ML Systems | Llama-3.1-8B | 3,132 | 3,078 | 2,817 | |
| | Qwen-2.5-7B | 4,343 | 4,274 | 3,459 | 9,493 |
| | DeepSeek-R1-7B | 4,278 | 4,096 | 3,414 | |
| Prob. Methods | Llama-3.1-8B | 2,676 | 2,647 | 2,208 | |
| | Qwen-2.5-7B | 4,397 | 4,329 | 4,046 | 9,373 |
| | DeepSeek-R1-7B | 3,995 | 3,852 | 3,253 | |

*Table 10.* Human calibration of the LLM audit stage ($N = 96$). $\kappa$: Cohen's $\kappa$; FP: false positives (LLM approves, human rejects); FN: false negatives (LLM rejects, human would accept).

| Domain | $N$ | Agreement | $\kappa$ | Precision | FP | FN |
|---|---|---|---|---|---|---|
| Overall | 96 | 89.6% | 0.79 | 97.9% | 2.1% | 18.8% |
| Deep Learning | 24 | 91.7% | 0.83 | 100% | 0% | 16.7% |
| ML Systems | 24 | 95.8% | 0.92 | 100% | 0% | 8.3% |
| Prob. Methods | 24 | 83.3% | 0.67 | 91.7% | 8.3% | 25.0% |
| Transformer Arch. | 24 | 87.5% | 0.75 | 100% | 0% | 25.0% |

*Table 11.* Representative atoms at each Bloom's taxonomy boundary.

| Type | Accepted Example | Rejected Example |
|---|---|---|
| Factual | "The Adam optimizer uses $\beta_1 = 0.9$, $\beta_2 = 0.999$, and $\epsilon = 10^{-8}$ as default hyperparameters." | "Deep learning uses optimizers." (too generic) |
| Conceptual | "Attention enables selective focus by computing a weighted combination of values, where weights are determined by query–key similarity." | "This is an important concept in NLP." (meta-statement) |
| Procedural | "Gradient clipping rescales the gradient when its $L_2$ norm exceeds a threshold $\theta$: $g \leftarrow g \cdot \theta / \|g\|_2$." | "You should clip gradients during training." (advice fragment, not a complete procedure) |

