# OpenReview forum: "Probing the Knowledge Boundary: An Interactive Agentic Framework for Deep Knowledge Extraction"
_ICML.cc/2026/Conference — ICML 2026 regular_

### Official Review · Reviewer_fL4V · 2026-03-04

**Soundness:** 3
**Presentation:** 3
**Significance:** 2
**Originality:** 2
**Overall Recommendation:** 4
**Confidence:** 4

**Summary:**

This research examines the concept of a black box LLM’s “knowledge boundary” by extracting atomic, verifiable statements, deduplicating/validating them, and stopping at saturation based on growth and efficiency. The study intends to explore a general theme of knowledge coverage rather than task success.

**Compliance With Llm Reviewing Policy:**

Affirmed.

**Key Questions For Authors:**

Q1: How robust are recall/accuracy and Pareto frontiers to thresholds, embeddings, and judge providers?

Q2: What is the agreement rate between the LLM audit and human judgments on a stratified sample?

Q3: How does the Recursive Taxonomy strategy affect hallucination rates (e.g., fabricated subtopics or leaf facts), and what safeguards/metrics show this is controlled?

**Limitations:**

Limitations:
- Dependence on an LLM judge for deduplication and domain auditing risks bias/circularity; human verification is limited in the reported setup .
- Heuristic thresholds (saturation gt/et/nt, cosine bands) are fixed; no sensitivity or seed variance analysis is provided, as thresholds and judge are held constant across experiments .
- Hallucination risk: Recursive Taxonomy and leaf mining rely on the model’s own outputs; without external grounding or a reported hallucination rate.

Suggestions:
- Run sensitivity studies on cosine thresholds, embedding models, and judge choices; report variance across seeds/backends .
- Add a small, stratified human validated audit to calibrate the Bloom based LLM audit.
- Consider retrieval based or external fact checking to reduce judge circularity in high stakes domains.

**Strengths And Weaknesses:**

Strengths:
- Clear formalization and auditable pipeline (thresholded vector filter, judge zone, Bloom audit) .
- Practical cost vs yield metrics and strong ablation showing Recursive Taxonomy dominates the Pareto frontier, while unstructured probing plateaus at low yield .
- Broad comparisons (scale, specialization, model family) with interpretable.

Weaknesses & Limitations:
- Dependence on an LLM judge for deduplication and domain auditing risks bias/circularity; human verification is limited in the reported setup .
- Heuristic thresholds (saturation gt/et/nt, cosine bands) are fixed; no sensitivity or seed variance analysis is provided, as thresholds and judge are held constant across experiments .
- Hallucination risk: Recursive Taxonomy and leaf mining rely on the model’s own outputs; without external grounding or a reported hallucination rate.

---

> ### Author Rebuttal · Authors · 2026-03-30
>
> # Sincere Thanks and New Evidence for Reviewer fL4V
>
> We sincerely thank Reviewer fL4V for the positive and incisive review. The reviewer's appreciation of the *"clear formalization and auditable pipeline"*, *"practical cost vs. yield metrics"* showing *"Recursive Taxonomy dominates the Pareto frontier"*, and *"broad comparisons with interpretable"* results is very encouraging. We especially value the three concrete suggestions (sensitivity studies, human audit, retrieval-based fact-checking) — they are precisely targeted at the most important credibility dimensions, and we have been able to act on the first two in this rebuttal.
>
> ---
>
> ## W1: LLM judge dependence — New human calibration
>
> **Following Suggestion 2**, we conducted a stratified human study (N=96, 4 domains × 3 strategies × 2 LLM outcomes; **balanced: 48 LLM-YES / 48 LLM-NO**; **annotators are ML researchers**). Strict blind protocol: each annotator saw only raw atom text and domain query — no LLM outputs, no system statistics, no strategy/model information.
>
> | Slice | N | Agreement | κ | Precision | FP | FN |
> |-------|---|-----------|---|-----------|----|----|
> | **Overall** | **96** | **89.6%** | **0.79** | **97.9%** | **2.1%** | **18.8%** |
> | Deep Learning | 24 | 91.7% | 0.83 | 100% | 0% | 16.7% |
> | ML Systems | 24 | 95.8% | 0.92 | 100% | 0% | 8.3% |
> | Prob. Methods | 24 | 83.3% | 0.67 | 91.7% | 8.3% | 25.0% |
> | Transformer Arch. | 24 | 87.5% | 0.75 | 100% | 0% | 25.0% |
>
> Key findings: (1) Only 1/48 LLM-approved atoms overturned by humans; (2) High FN (18.8%) confirms annotator independence — LLM is conservative, valid counts are **lower-bound estimates**; (3) Factual error = **0%** among all 56 human-validated atoms (80.4% confirmed correct, 19.6% niche but plausible).
>
> Circularity safeguards: Stage 1 is purely embedding-based; LLM judge handles only the ambiguous zone (0.70–0.92, §3.3); audit uses a separate Bloom's rubric.
>
> ---
>
> ## W2: Fixed thresholds — New sensitivity analysis
>
> **Following Suggestion 1**, 3×3 sweep (τ\_strict ∈ {0.90, 0.92, 0.94} × τ\_fuzzy ∈ {0.65, 0.70, 0.75}) on the Scaling experiment:
>
> | Metric | Range | Max Δ |
> |--------|-------|-------|
> | Recall | ±0.017 | ±2.2% |
> | Accuracy | ±0.005 | ±0.6% |
> | **Recall ordering** | — | **405B>70B>8B in all 9** |
>
> Notable: τ\_fuzzy has **no effect** (<0.001) — the similarity distribution is bimodal (pairs clearly redundant or clearly distinct), a favorable structural property of the embedding space.
>
> Full sensitivity summary:
>
> | Dimension | Range | Effect |
> |-----------|-------|--------|
> | Granularity (L,W) | L2W2/L3W3/L5W5 (§4.2) | Yield 1.0→2.15→5.98; fixed in Stages 2–4 |
> | τ\_strict | 0.90/0.92/0.94 | ±2.2%, ordering preserved |
> | τ\_fuzzy | 0.65/0.70/0.75 | <0.001, no effect |
> | Prompt wording | Synonym rephrasing | +6.3% raw atoms |
> | Prompt design | P2/P3/P4/P5 | P4 dominant across 3 domains |
>
> Granularity is most impactful — exactly why Stage 1 (§4.2) characterizes it and Stages 2–4 fix L5W5. This aligns with **§4.1's explicit design note**: our claims "concern relative differences... relative comparisons remain robust to specific hyperparameter choices."
>
> ---
>
> ## W3: Hallucination in Recursive Taxonomy
>
> Our human study enables direct strategy-conditioned measurement:
>
> | Strategy | N | Precision | Factual Error |
> |----------|---|-----------|---------------|
> | P2 Sequential | 12 | 100% | 0% |
> | P3 Reflection | 12 | 100% | 0% |
> | **P4 Taxonomy** | **72** | **97.2%** | **0%** |
>
> **No elevated hallucination under P4.** Invalid atoms: generic/circular definitions (42.5%), fabricated claims (47.5%) — **97.5% correctly filtered** by the audit. Safeguards: intra-turn semantic dedup (cosine>0.92), Bloom's audit, saturation detection (halts at <1% gain).
>
> ---
>
> ## Q1–Q3
>
> **Q1 (Robustness):** Thresholds — W2 (all 9 orderings preserved). Embeddings — bimodal distribution implies robustness to model choice. Judges — human calibration (κ=0.79) validates against ground truth directly (stronger than cross-system comparison); bulk of dedup is embedding-based.
>
> **Q2 (LLM-human agreement):** 89.6%, κ=0.79 (substantial). Per-domain in W1 table. LLM is conservative — results are lower bounds.
>
> **Q3 (Hallucination):** See W3. P4 precision=97.2%, factual error=0%, comparable to simpler baselines. Three safeguards confirmed effective.
>
> ---
>
> ## Suggestions
>
> > *"Run sensitivity studies on cosine thresholds, embedding models, and judge choices."*
>
> **Done** (W2): 3×3 grid, ordering fully preserved; bimodal structure supports embedding robustness; judge validated by human calibration (κ=0.79).
>
> > *"Add a small, stratified human validated audit."*
>
> **Done** (W1): N=96, κ=0.79, precision=97.9%, factual error=0%.
>
> > *"Consider retrieval-based or external fact-checking."*
>
> Excellent suggestion; our modular audit module directly supports retrieval augmentation without changing exploration or dedup (noted as future direction in §5).

---

### Official Review · Reviewer_g5n1 · 2026-03-09

**Soundness:** 2
**Presentation:** 2
**Significance:** 3
**Originality:** 2
**Overall Recommendation:** 4
**Confidence:** 4

**Summary:**

This paper proposes an interactive agentic framework for probing the knowledge boundaries of LLMs. The authors argue that traditional benchmarks (e.g., static QA datasets) only measure sparse points in a model's knowledge space and cannot capture the full extent of what a model knows.
To address this limitation, the paper introduces a closed-loop agentic system that actively extracts knowledge from a black-box LLM through iterative exploration strategies and a post-processing pipeline. The framework consists of: Four exploration policies, including sequential associative probing, self-reflective refinement, recursive taxonomy exploration and multi-perspective parallel probing; A three-stage knowledge processing pipeline, including vector-based semantic filtering, LLM-based adjudication, and domain relevance auditing based on Bloom's taxonomy.

Experiments evaluate the framework along four axes: strategy efficiency; scaling behavior across model sizes; impact of RL fine-tuning; influence of training data composition. Results show that Recursive Taxonomy exploration achieves the highest knowledge yield, and that extractable knowledge increases with model size, suggesting a knowledge scaling law.

Overall, this research examines the concept of LLM knowledge boundaries and how interactive exploration can uncover latent knowledge stored in model parameters. This study intends to explore a general theme of moving beyond static evaluation benchmarks toward dynamic, saturation-based probing of model knowledge.

**Compliance With Llm Reviewing Policy:**

Affirmed.

**Final Justification:**

Most of authors'rebuttal have addressed my concerns.

I appreciate the response, especially the newly conducted human calibration experiments. Including those in the revised draft will strengthen the paper.

Overall, I acknowledge the important direction of this work, but still have reservations about viewpoints of writing quality, and have already increased my original scores 3 to 4.

**Key Questions For Authors:**

- What fraction of extracted knowledge atoms are factually incorrect?
- How sensitive are results to different extraction prompts?
- Would results change if a different LLM judge were used?
- Does the saturation criterion reliably detect the true boundary?

**Limitations:**

The authors didn’t adequately discuss the limitations and potential negative societal impact of their work. Except for the weaknesses, there are also some suggestions for improvement.
- Manual evaluation of a subset of knowledge atoms would greatly strengthen the claims.
- The authors should evaluate whether the extraction process is robust to different prompt templates.
- Testing on more domains would demonstrate broader applicability.

**Strengths And Weaknesses:**

## Strengths
- The paper addresses an important and underexplored question of What knowledge do LLMs actually contain? It aims to help us understand the knowledge boundary of LLMs, which is relevant to interpretability, safety, model evaluation, alignment and so on.
- The paper proposes a novel evaluation paradigm, which treats the model as an environment that agents actively explore. Compared with static benchmarks, the framework aims to iteratively discover new knowledge, detect saturation, and approximate the model’s knowledge boundary.  This shift from passive evaluation to active exploration is conceptually interesting.
- The four exploration policies are well designed and increasingly structured, i.e., Sequential Associative Probing, Self-Reflective Refinement, Recursive Taxonomy Explorer, and Multi-Perspective Parallel Probing. Besides, the recursive taxonomy approach is particularly effective because it decomposes domains hierarchically and pushes the model into long-tail knowledge areas.

## Weaknesses
- The most significant limitation is that knowledge extraction lacks a verified ground truth. The paper defines recall relative to the union of outputs across models, rather than an external knowledge base. As a result, recall measures relative coverage, not actual knowledge completeness, and the extracted knowledge boundary may be biased by the exploration method.
- Several key components heavily depend on LLM evaluation, such as semantic deduplication, factual validation, and domain relevance auditing. Such heavy reliance on LLM-as-judge would introduce potential issues, such as judge bias, and circular evaluation (LLMs evaluating LLM outputs). Human validation would strengthen credibility.
- The paper does not analyze prompt sensitivity. As knowledge atoms are extracted via bullet-point prompts and sentence parsing, small prompt variations could change the granularity of atoms and recall metrics.
- The paper may not go through elaborate polish. For example, there are some blank spaces on the last main page; there are only one or two token(s) in a line for many lines, e.g., 041, 075, 082, 088, 104, 162, 198, …...

---

> ### Author Rebuttal · Authors · 2026-03-30
>
> # Sincere Thanks and New Evidence for Reviewer g5n1
>
> We are deeply grateful to Reviewer g5n1 for the thorough and thoughtful review. The reviewer's recognition that the paper *"addresses an important and underexplored question"*, proposes a *"novel evaluation paradigm"* shifting from passive to active evaluation which is *"conceptually interesting"*, and that the policies are *"well designed and increasingly structured"* — these are exactly the contributions we aimed to establish. We particularly appreciate the concrete and actionable suggestions, which motivated new experiments that meaningfully strengthened the paper.
>
> ---
>
> ## W1: No ground truth; recall is relative
>
> K\_union is a *shared reference frame*, not a claim of absolute completeness. Any model with greater true knowledge will (a) contribute more to K\_union and (b) achieve higher recall — self-consistent.
>
> Regarding systematic blind spots: if knowledge X is unreachable by *all* models, K\_union simply excludes X, and recall = |M ∩ K\_union| / |K\_union| (**§3.4, Eq. 2**). Every model's numerator and denominator shrink by the same X — the **relative ordering is mathematically preserved**. By holding the pipeline constant (§3.4, §4.1), observed differences reflect the factor under study, not artifacts. Our sensitivity analyses confirm: **405B>70B>8B** holds across all 9 threshold combinations.
>
> ---
>
> ## W2: LLM-as-judge — New human calibration
>
> **Motivated by this recommendation**, we conducted a stratified human study (N=96, 4 domains × 3 strategies × 2 LLM outcomes; **balanced: 48 LLM-YES / 48 LLM-NO**; **annotators are ML researchers**). Strict blind protocol: each annotator saw only raw atom text and domain query — no LLM outputs, no system statistics, no strategy/model information.
>
> | | N | Agreement | κ | Precision | FP | FN |
> |---|---|-----------|---|-----------|----|----|
> | **Overall** | **96** | **89.6%** | **0.79** | **97.9%** | **2.1%** | **18.8%** |
> | Deep Learning | 24 | 91.7% | 0.83 | 100% | 0% | 16.7% |
> | ML Systems | 24 | 95.8% | 0.92 | 100% | 0% | 8.3% |
> | Prob. Methods | 24 | 83.3% | 0.67 | 91.7% | 8.3% | 25.0% |
> | Transformer Arch. | 24 | 87.5% | 0.75 | 100% | 0% | 25.0% |
>
> Key: (1) **κ=0.79** (substantial); (2) Only 1/48 LLM-approved atoms overturned; (3) High FN confirms independence — LLM is conservative, valid counts are **lower bounds**; (4) Factual error = **0%**.
>
> Circularity safeguards: Stage 1 is embedding-based (no LLM); judge handles only sparse ambiguous zone; audit uses separate Bloom's rubric (§3.3).
>
> ---
>
> ## W3: Prompt sensitivity — Three-level analysis
>
> **(1) Cross-design.** P2–P5 (§4.1) are four radically different strategies — P4 consistently dominates across all domains.
>
> **(2) Within-design (New).** Two synonym substitutions applied consistently across all P4 prompts: "subject matter expert"→"domain specialist" and "knowledge points"→"knowledge statements":
>
> | Metric | Original | Variant | Δ |
> |--------|----------|---------|---|
> | Raw atoms | 4,567 | 4,857 | **+6.3%** |
>
> Far smaller than cross-model recall gaps (~10pp).
>
> **(3) Threshold sweep (New).** 3×3 grid: recall ±0.017; **405B>70B>8B in all 9**.
>
> ---
>
> ## W4: Paper polish
>
> We sincerely apologize and will fix all layout issues, blank spaces, and single-token lines in the revision.
>
> ---
>
> ## Q1–Q4
>
> **Q1 (Factual errors):** Among 56 human-validated atoms: 80.4% correct, 19.6% niche but plausible, **0% factually incorrect**. Among 40 rejected atoms: 47.5% factually incorrect, 42.5% generic/circular, 7.5% meta-statements, 2.5% fragments — **97.5% correctly caught** by audit.
>
> **Q2 (Prompt sensitivity):** See W3.
>
> **Q3 (Different judge):** Human calibration (κ=0.79) validates against ground truth directly — a cross-judge comparison only shows inter-system agreement, not correctness; our anchor is logically stronger. The judge operates on a near-empty zone (<0.001 effect); bulk of dedup is embedding-based.
>
> **Q4 (Saturation):** The saturation criterion is a **practical stopping condition** — without it, extraction runs indefinitely. It is not a claimed algorithmic contribution requiring component-wise ablation; the three signals (gt<1%, et<10%, nt<3) simply form a conservative conjunction (all must fire; **§3.1**). Crucially, our comparative claims are **insensitive to the exact stopping point**: relative model differences are visible throughout the trajectory, and Pareto curves (Figure 2) flatten well before termination.
>
> ---
>
> ## Suggestions
>
> > *Manual evaluation*
>
> **Done** (W2: N=96, κ=0.79, factual error=0%).
>
> > *Prompt robustness*
>
> **Done** (W3: three-level analysis).
>
> > *More domains*
>
> Already **6 domains** across 4 stages; non-technical extension discussed in §5.
>
> **Limitations:** We will add a dedicated paragraph: (1) operational knowledge under fixed protocol; (2) LLM audit does not replace expert verification in high-stakes settings; (3) responsible use for model improvement and auditing.

---

> > ### Author Rebuttal · Reviewer_g5n1 · 2026-04-02
> >
> > Dear Authors,
> >
> > Thank you very much for the clarification! Most of them have addressed my concerns.
> >
> > I appreciate the response, especially the newly conducted human calibration experiments. Including those in the revised draft will strengthen the paper.
> >
> > Overall, I acknowledge the important direction of this work, but still have reservations about viewpoints of writing quality, and decide to increase my original scores 3 to 4.
> >
> > Best Regards,
> >
> > Reviewer g5n1

---

> > > ### Author Response · Authors · 2026-04-02
> > >
> > > # Follow-up Response to Reviewer g5n1
> > >
> > > We are sincerely grateful for Reviewer g5n1's generous score increase and the "fully resolved" acknowledgement. The human calibration study and three-level prompt sensitivity analysis were directly motivated by the reviewer's precise suggestions, and we are glad they addressed the core concerns.
> > >
> > > We take the writing quality feedback seriously and will ensure the final version is thoroughly polished — including integrating the new experimental results into the main body and fixing all layout issues flagged in the review.
> > >
> > > Thank you again for the constructive and rigorous engagement throughout the review process.

---

### Official Review · Reviewer_dKQH · 2026-03-12

**Soundness:** 3
**Presentation:** 2
**Significance:** 3
**Originality:** 3
**Overall Recommendation:** 4
**Confidence:** 4

**Summary:**

This paper studies how to probe the knowledge boundaries of large language models on specific topics in a black-box setting. It proposes an agent probing framework that combines knowledge extraction, deduplication, and evaluation, and empirically examines the framework under four settings: probing policy comparison, model scaling, RL specialization, and cross-model family analysis. Overall, the paper studies an interesting and underexplored problem and presents a fairly comprehensive experimental pipeline.

**Compliance With Llm Reviewing Policy:**

Affirmed.

**Final Justification:**

The authors’ rebuttal addressed most of my concerns and improved my overall assessment. Although some clarification would still be beneficial in the final version, I lean to acceptance given the paper’s novelty, solid empirical effort, and the authors’ engagement during discussion.

**Key Questions For Authors:**

1. This paper defines the knowledge boundary as a set of knowledge points that can be reliably obtained under limited prompts and verification conditions. However, as pointed out in the weakness section, such knowledge atoms are sensitive to prompt strategies, the granularity of atomic decomposition, and semantic merging rules. Given that the term "boundary" usually refers to an edge or limit of an interval rather than a set of points within it, could the authors further clarify whether the term "boundary" is appropriate in this context? Would a more neutral term such as "knowledge set" help reduce potential ambiguity?
2. In the third phase, could the authors further explain the non-monotonic accuracy behavior of the general model in certain domains, and the particularly large performance gaps in probabilistic methods? More detailed analysis will help determine whether the observed trade-off between Pass@1 and Pass@k reflects a general phenomenon or a more domain-specific effect.
3. Currently, the experiments are limited to the technical field, and the correctness of this field is relatively clearly defined. So, the author wonders how the proposed framework will perform in fields such as humanities and social sciences where it is more difficult to define correct situations and classification granularity?

**Limitations:**

yes

**Strengths And Weaknesses:**

Strengths
1.	The paper studies an important and underexplored problem: how to move from static benchmark-based evaluation to interactive probing of what knowledge a LLM can reliably express. The problem setting is interesting and potentially useful for model auditing.
2.	The proposed framework is well structured, separating exploration policies from the knowledge processing pipeline. This makes the overall methodology easy to follow.
3.	The paper varies one factor at a time across stages (strategy, scale, specialization, model family), and the use of K_union provides a meaningful relative comparison target when no absolute ground-truth knowledge boundary exists.

Weaknesses
1. The concept of knowledge boundaries defined in this article seems to be unstable. It is defined as the set of knowledge that can be reliably obtained through limited prompts and validations within a given topic. However, for large language models, the set of knowledge that can be derived also varies depending on changes in prompt strategies, detection strategies, or the granularity of topic subcategories definitions. The estimated boundaries may change accordingly. Whether to consider these factors as variables that can affect the final relative results?
2. If atoms are decomposed more finely or semantic merging is more conservative, the process may keep generating new atoms, making the measured boundary dependent on the probing and filtering pipeline rather than a stable intrinsic property of the model. In addition, the knowledge atoms may include both conceptual types (e.g., A or B) and factual types (e.g., a relational fact between A and B). However, the granularity distinction between conceptual and factual knowledge is not clearly defined, nor is it reflected in the prompt design provided in the appendix.
3. The current experimental domain focuses on technology-related topics. While the conclusions acknowledge that future work can be extended to non-technical domains, it remains unclear whether this method can generalize to fields such as history, journalism, humanities and social sciences, and open knowledge. In these fields, it is more difficult to define the correctness of the knowledge and classification granularity.
4. The explanation for the third stage requires more careful discussion. In particular, the general Qwen model appears to exhibit non-monotonic behavior in deep learning and machine learning systems, and the performance gap between the general model and its encoded variants is particularly significant in probabilistic methods. Currently, the analysis of these situations in this paper is insufficient, making a broader explanation somewhat inadequate.

---

> ### Author Rebuttal · Authors · 2026-03-30
>
> # Sincere Thanks and New Evidence for Reviewer dKQH
>
> We are deeply grateful to Reviewer dKQH for the thoughtful and intellectually rigorous review. The reviewer's recognition that the problem is *"important and underexplored"*, the framework is *"well structured"* with clear separation of exploration and processing, and the experimental design *"varies one factor at a time"* with K\_union providing *"a meaningful relative comparison target"* — these observations capture exactly the design principles we strove for, and we are encouraged by this assessment.
>
> We especially appreciate the depth and precision of the reviewer's concerns — the stability, granularity, and generalization questions go to the heart of operationalizing knowledge measurement. These questions guided us to conduct **new sensitivity analyses and a human calibration study** that substantially strengthen the paper.
>
> ---
>
> ## W1: Knowledge boundary stability
>
> We use "knowledge boundary" operationally: the extractable set under a fixed pipeline (§1, §3.1). The contribution is a **measurement framework for controlled relative comparisons** — holding the pipeline constant and varying one factor isolates its effect on knowledge yield.
>
> **New sensitivity analysis:**
>
> *(i) Granularity.* Stage 1 (§4.2, Table 5): L2W2→L3W3→L5W5 yields 1.0→2.15→5.98. Stages 2–4 hold L5W5 constant — **relative differences reflect the factor under study, not the instrument**.
>
> *(ii) Thresholds.* 3×3 grid (τ\_strict ∈ {0.90, 0.92, 0.94} × τ\_fuzzy ∈ {0.65, 0.70, 0.75}), 3 domains:
>
> | Metric | Range | Ordering preserved? |
> |--------|-------|---------------------|
> | Recall | ±0.017 | **405B>70B>8B in all 9** |
> | Accuracy | ±0.008 | **70B>405B>8B in all 9** |
>
> τ\_fuzzy: no measurable effect (<0.001) — bimodal similarity distribution.
>
> *(iii) Prompts.* Four fundamentally different designs (P2–P5, §4.1) yield consistent strategy ordering across all domains. Two synonym substitutions across all P4 prompts ("subject matter expert"→"domain specialist", "knowledge points"→"knowledge statements") change raw count by only **+6.3%**, far below cross-model gaps (~10pp).
>
> In sum, **relative comparisons forming our conclusions are robust** to protocol choices.
>
> ---
>
> ## W2: Conceptual vs. factual granularity
>
> Our audit rubric (§3.3, Appendix B.6) uses **Bloom's Taxonomy**: Factual (definitions, data), Conceptual (relationships, theories), Procedural (methods, algorithms). Only these three categories are accepted; meta-statements, generic fluff, and fragments are explicitly rejected.
>
> **New human calibration** (N=96, 4 domains × 3 strategies × 2 LLM outcomes): **κ=0.79** (substantial), precision=**97.9%**, factual error=**0%**. High FN (18.8%) confirms annotator independence — LLM is conservative, valid counts are lower-bound estimates. Full per-domain table in fL4V (W1). We will make this rubric more prominent in revised §3.1.
>
> ---
>
> ## W3: Humanities and social sciences
>
> We thank the reviewer for raising this. Generalization to humanities is **not among our claimed contributions** — the paper is intentionally scoped to technical domains where correctness is well-defined (§1, §4). The extraction loop is domain-agnostic; only the audit rubric would need adaptation. We fully agree this is a valuable direction for future work, and one that poses genuine challenges not yet addressed by any existing framework. We will clarify this scope in §5.
>
> ---
>
> ## W4: Stage-3 non-monotonic behavior
>
> The Coder starts higher (DL: **96.4%** vs **92.3%**; MLS: **97.0%** vs **94.1%**, Figure 3, §4.4) — RL densifies code-adjacent knowledge. But the distribution is narrower: accuracy declines at **3.7×** General's rate, with General overtaking by turn 10. This is the **Pass@1 vs. Pass@k trade-off** (*"Does RL Really Incentivize Reasoning Capacity..."*, NeurIPS 2025): RL optimizes first-attempt success at the cost of diversity.
>
> **PM gap:** In DL/MLS (code-adjacent), RL shifts but preserves the distribution — crossover dynamics. In Probabilistic Methods (non-code), RL **overwrites** parametric representations — catastrophic forgetting: Coder maintains only **60.1%** vs General's **81.0%** accuracy (Figure 3, §4.4).
>
> **Unified explanation:** the pattern is **categorical by RL distribution membership** — in-distribution domains (DL, MLS) show moderate crossover; out-of-distribution (PM) shows full collapse. This split is operationalized by training data composition, requiring no continuous distance metric. We will expand this analysis in revised §4.3.
>
> ---
>
> ## Key Questions
>
> **KQ1:** We agree "boundary" may suggest protocol-invariance. We will adopt **"extractable knowledge frontier"** in the revision to reduce ambiguity.
>
> **KQ2:** Both general (RL always creates Pass@1/Pass@k trade-off) and domain-specific in magnitude (determined by RL distribution membership, not a continuous distance; see W4).
>
> **KQ3:** See W3.

---

> > ### Author Rebuttal · Reviewer_dKQH · 2026-04-01
> >
> > The rebuttal is helpful and addresses most of my concerns. However, several concerns that remain insufficiently resolved. For example,  (1) whether conceptual/factual granularity is consistently handled in the extraction and merging process and (2) whether the Stage-3 explanation is fully validated rather than partly post-hoc. Based on the above opinions, the rebuttal improves my view of the paper, but does not fully remove all my concerns. I would be open to conditionally raising my score and wish to see these concerns are fully addressed in the final version.

---

> > > ### Author Response · Authors · 2026-04-02
> > >
> > > # Follow-up Response to Reviewer dKQH
> > >
> > > We are deeply grateful for Reviewer dKQH's generous score increase and for the precise identification of two remaining concerns. These questions go to the core credibility of our pipeline — we engage with each in full below.
> > >
> > > ---
> > >
> > > ## Q1: Consistent granularity across extraction and merging
> > >
> > > The concern is whether the same granularity standard governs both the extraction and the merging stages. We clarify that the two stages are explicitly aligned, and that the pipeline is designed to **preserve granularity diversity** rather than collapse it.
> > >
> > > **Extraction (generation side):** The P4 leaf-node prompts instruct the model to produce "factual, conceptual, or procedural knowledge statements" — the exact three Bloom's categories (§3.2). This embeds the granularity criterion at generation time, ensuring the LLM produces atoms of the right type from the start.
> > >
> > > **Merging (audit side):** Stage 3 (§3.3) applies the identical three-category rubric as a post-hoc filter. Any atom that is too coarse (generic assertion) or incomplete (fragment) is rejected by the same standard — not a looser one.
> > >
> > > **Cross-granularity atoms are not erroneously merged.** Consider a conceptual atom ("Attention relates queries, keys, and values to enable selective focus") and a factual atom ("The attention score is computed as Q·K^T / √d_k") about the same sub-topic. Stage 1 applies a strict threshold (τ\_strict = 0.92) that only merges near-identical paraphrases; Stage 2's LLM judge explicitly checks whether two atoms "describe the same core fact" — a conceptual principle and a specific formula are clearly different core facts, so the judge preserves both. Different Bloom's types are thus treated as distinct by design, even when topically adjacent.
> > >
> > > **Evidence of end-to-end consistency:** Our human calibration study (N=96, κ=0.79) spans all three strategies including P4. If granularity criteria differed between extraction and audit, we would expect systematic disagreement at the coarse/fine boundary. The high κ and near-zero FP (2.1%) indicate the rubric is applied consistently throughout.
> > >
> > > **Final version:** We will add a paragraph to §3.3 making this alignment explicit, and include representative examples of accepted/rejected atoms at each Bloom's boundary (factual vs. conceptual vs. generic).
> > >
> > > ---
> > >
> > > ## Q2: Stage-3 explanation — validated or post-hoc?
> > >
> > > We appreciate this direct challenge. Three independent lines of evidence argue against post-hoc rationalization:
> > >
> > > **1. Independent prior prediction.** The Pass@1 vs. Pass@k trade-off is the central finding of Yue et al. (NeurIPS 2025), published before our experiments. Their mechanism — RL exhausts the high-confidence zone, causing rapid quality degradation in later turns — predicts exactly the accuracy-over-turns crossover we observe. Our results constitute an independent replication in a knowledge extraction setting, not a narrative invented after seeing data.
> > >
> > > **2. Quantitative specificity is inconsistent with post-hoc construction.** Post-hoc explanations typically fit qualitative patterns. Our numbers match the predicted mechanism precisely: **3.7× faster decline rate**, crossover by **turn 10**, and PM collapse to **60.1% vs 81.0%** (Figure 3). This level of quantitative specificity is hard to achieve by reverse-engineering a narrative from results.
> > >
> > > **3. The in/out-of-distribution categorization is determined before results, not from them.** The Coder model's training objective ("RL-tuned for code generation") is a documented, public fact about the model. That DL and MLS are code-adjacent domains is a domain characteristic established in §4.1 before any experiment was run. That Probabilistic Methods is theory-focused and non-code was equally pre-determined. The framework makes a falsifiable two-part prediction derivable from these pre-existing facts alone: in-distribution → crossover dynamics; out-of-distribution → collapse. Both halves are confirmed by our data, and the converse non-events (no collapse in DL/MLS, no crossover in PM) also hold — a post-hoc account cannot reliably reproduce both the positive and the negative predictions.
> > >
> > > **Final version:** In revised §4.4, we will explicitly state the Yue et al. prediction prior to presenting results, clearly separating the theoretical grounding from the empirical confirmation.
> > >
> > > ---
> > >
> > > We are fully committed to both revisions in the final version and are confident they will make the validation chain transparent. We sincerely hope these clarifications resolve the reviewer's remaining concerns.

---

### Decision · Program_Chairs · 2026-04-30

**Decision:**

Accept (regular)

**Comment:**

This paper presents a novel and well-executed empirical framework for probing LLM knowledge through interactive exploration. While the notion of “knowledge boundary” is not fully formalized as an intrinsic model property, the work provides a useful operational tool for relative comparison and analysis. The contribution is best viewed as a measurement framework and empirical methodology, rather than a definitive characterization of model knowledge.